# Association Between Current Suicidal Ideation and Personality Traits: Analysis of the Personality Inventory for DSM-5 in a Community Mental Health Sample

**DOI:** 10.3390/medicina61091541

**Published:** 2025-08-27

**Authors:** Valeria Deiana, Pasquale Paribello, Federico Suprani, Andrea Mura, Carlo Arzedi, Mario Garzilli, Laura Arru, Mirko Manchia, Bernardo Carpiniello, Federica Pinna

**Affiliations:** 1Unit of Psychiatry, Department of Medical Sciences and Public Health, University of Cagliari, 09127 Cagliari, Italy; v.deiana83@gmail.com (V.D.); pasquale.paribello@unica.it (P.P.); federicosuprani@hotmail.it (F.S.); andremura88@gmail.com (A.M.); carloarzedi@yahoo.it (C.A.); m.garzi@gmail.com (M.G.); laura.arru282@gmail.com (L.A.); bcarpini@iol.it (B.C.); federica.pinna@unica.it (F.P.); 2Unit of Clinical Psychiatry, University Hospital Agency of Cagliari, 09127 Cagliari, Italy; 3Department of Pharmacology, Dalhousie University, Halifax, NS B3H4R2, Canada

**Keywords:** suicide behavior, risk prevention, personality traits, trauma, depression

## Abstract

*Background and Objectives*: Suicide is one of the leading causes of death worldwide. Identifying psychopathological factors and personality traits associated with suicidal ideation is crucial for improving prevention. This study investigates the association between current suicidal ideation—measured by item 4 of the Brief Psychiatric Rating Scale-24 items (BPRS-24)—and personality traits assessed with the Personality Inventory for DSM-5 (PID-5) in a transdiagnostic outpatient psychiatric population. The association of BPRS-24 item 4 with early traumatic experiences, attachment styles, and dissociative phenomena is also explored as a secondary objective. *Materials and Methods*: We propose a secondary analysis on a sample of 137 individuals receiving care at an academic community mental health center. Personality traits were assessed using the PID-5, while attachment styles were assessed with the Experiences in Close Relationships-Revised (ERC-R), childhood traumas with the Childhood Experience of Care and Abuse Questionnaire (CECA.Q), and dissociative experiences with the Dissociative Experience Scale II (DES-II). Suicidal ideation was measured with item 4 of the BPRS-24. Associations were examined using Spearman’s correlation and ordinal logistic regression models, adjusted for age, sex assigned at birth, and global clinical severity (CGI-S). *Results*: We found statistically significant associations between suicidal ideation and the PID-5 trait of depressivity (OR = 1.80, 95 % CI 1.23–2.63, *p* = 0.002, *p*-value adjusted Holm’s method 0.012). However, this association lost significance after adjusting for depressive symptoms (BPRS-24 item 3), suggesting a mediating role of depression. We found no significant associations with childhood trauma, attachment styles, or dissociative experiences. *Conclusions*: Our findings suggest a potential link between specific personality traits and suicidal ideation, mediated by depressive symptomatology. We explore how future studies may evaluate PID-5 as a clinical tool to support the identification of individuals at long-term risk of suicidality or for targeting specific populations for tailored interventions.

## 1. Introduction

Suicide is one of the leading causes of death worldwide, with approximately 800,000 deaths each year [1]. Despite a relatively descending trend during the last decades, it still represents the 17th cause of death globally as of 2019, and poignantly, the mortality of suicide appears to increase in specific populations [2]. Death by suicide represents a negative outcome influenced by the complex interplay of multiple biological, psychological, and environmental factors (Figure 1) [3]. Discrete heterogeneity in the prevalence of suicide exists even within the same geographical region, depending on numerous factors such as ethnicity, demography, and overall health status [3]. Research efforts applying a reductionist approach to suicide failed to grasp the complexity of the interdependent interaction of multiple factors, typical of dynamical systems [4].

Moreover, a large body of research on suicide behaviour has focused on clinical populations receiving psychiatric care. However, it is increasingly recognized how death by suicide may indeed frequently occur among non-clinical populations, with only 20% of suicide decedents being engaged in psychiatric care [5]. Evidence from the violent death registry suggests that a large and growing proportion of individuals dying by suicide (31%) featured prominent physical health concerns and minimal psychotropic medication usage as compared with other suicide profiles [6]. Considering the foregoing and as postulated by Rosen in the early 20th century [7], no clinically viable prediction instrument for suicide has emerged as one would reasonably expect for rare outcomes, with prevention studies indicating small to moderate effects of efficacy [8,9]. Far from leading to clinical and research nihilism, this realization should prompt realistic expectations for what could be reasonably expected from this field of research. Specifically, it is plausible that studies focusing on suicidality in specific clinical populations at enriched risk could have particular relevance. For example, suicide-focused treatment, derived from mechanistic, empirical models of suicidality, may, in fact, represent a promising avenue for the future development of acceptable and efficacious interventions [10,11]. Dissociative experiences and life traumas have both been reported in association with an increased frequency of suicide ideation and attempts in clinical populations [12]. Similarly, personality traits, such as higher neuroticism and lower levels of extraversion, are associated with a higher probability for suicidal behaviors [13]. Conversely, hyperthymic temperament appears to be protective against suicide behavior [14,15]. Here, we describe the potential association of the Brief Psychiatric Rating Scale-24 items, which defines suicide ideation and attempts (BPRS-4), with a psychological profile comprising the Personality Inventory for DSM-5 (PID-5), dissociative experiences with the use of the Dissociative Experience Scale II (DES II), early-life traumas with the use of the Childhood Experience of Care and Abuse Questionnaire (CECA.Q), and attachment styles with the Experiences in Close Relationships-Revised (ECR-R) in a transdiagnostic sample of individuals receiving care from an outpatient mental health clinic. Furthermore, in this cross-sectional study, we aim to explore whether there might be evidence for the use of PID-5 as a clinical tool to support the identification of individuals featuring a higher clinician-rated BPRS-4 suicide risk.

## 2. Materials and Methods

We present a secondary analysis performed on a sample already described in a previous publication from our research group [16]. We recruited a sample comprising subjects receiving care at the Unit of Psychiatry of the Department of Medical Sciences and Public Health of the University of Cagliari in Italy from January 2014 to December 2017. This is a psychiatric outpatient clinic part of the National Healthcare System in Italy with a catchment area of approximately 70,000 subjects and nearly 2400 individuals regularly receiving care. Study participants were recruited through word of mouth and by directly asking the responsible physician for the subject’s suitability for the present study project. We excluded patients who were unable to fill in the questionnaire or interact constructively with the researchers involved in the present project due to cognitive problems or language barriers. Moreover, we excluded patients with a Clinical Global Impression scale—Severity higher than or equal to 5 during recruitment, suggesting significant symptom exacerbation for the underlying disorder(s). Relevant data for the present study were gathered through direct interviewing and analysis of the available medical health records.

### 2.1. Psychometric Measures

At recruitment, we gathered sociodemographic data and proposed a test panel including both self- and clinician-rated instruments as follows.

#### 2.1.1. Personality Inventory for DSM-5 (PID-5)

The PID-5 for adults is a 220-item questionnaire aimed at individuals ≥ 18 years old. It proposes a series of sentences and requires the subject to indicate how frequently each element appears true or false in their daily life by applying a score on a scale from 0 (always or frequently false) to 3 (always or frequently true). We calculated the corresponding domains and traits.

#### 2.1.2. Childhood Experience of Care and Abuse (CECA.Q)

The CECA.Q is a questionnaire version of the Childhood Experience of Care and Abuse Interview, covering parental loss, neglect, antipathy, and physical and sexual abuse before age 17 [17].

#### 2.1.3. Brief Psychiatric Rating Scale—24 Items (BPRS-24)

The BPRS-24 or BPRS-E [18] is an extended version of the original BPRS-16 [19], which has displayed good internal consistency and interrater reliability [20,21]. The BPRS-24 offers a semi-structured interview assessing a wide range of psychopathological elements, ranging from depression to thought disorders, and it is scored on a range from 1 (symptom absent) to 7 (highest severity). It has been shown to present decent inter-rater reliability, even among raters with high vs. low proficiency [22]. Particularly relevant to the present analysis is item 4, addressing suicide risk, which is defined as “Suicidal thoughts, expressed intention, self-harming behaviors, suicide attempts”. The way the corresponding anchor for each score is phrased may lead to attributing a score of 5 or higher for both individuals who have attempted suicide and those with active ideation and planning, somewhat conflating ideation with attempted suicide (Table 1).

#### 2.1.4. Dissociative Experience Scale II (DES-II)

The DES-II [23] is a 28-item self-report measure of dissociative experiences such as derealization, depersonalization, absorption, and amnesia [24]. Each item is assessed on a numbered scale from 0 to 100%, indicating never or always at each pole, respectively.

#### 2.1.5. Clinical Global Impression—Severity (CGI-S)

The CGI-S is a user-friendly, transdiagnostic measure applied in research and in clinical practice [25]. This can be aptly applied for an initial assessment and tracking of clinical outcomes in treating the most varied clinical conditions. It typically requires a minimal amount of time for scoring. It allows for a readily understandable severity measure by indicating clinical severity on a seven-point scale, ranging from 1 to 7. Due to its nature, extensive experience is required on the part of the clinician in evaluating the overall severity of the considered case, encompassing the relevant clinical history, current psychosocial circumstances, behavior, and impact on functioning [25].

#### 2.1.6. Experiences in Close Relationships-Revised (ECR-R)

The ECR-R is a 36-item questionnaire measuring how individuals behave in relations with others, and it comprises two subscales of attachment, avoidance and anxiety. Avoidant attachment is defined as a pattern of discomfort with intimacy and seeking in the context of affective relations, whilst individuals featuring anxious attachment tend to fear abandonment and rejection [26]. Each item is scored on a scale from 1 to 7, corresponding to high agreement or high disagreement at each extremity, respectively.

#### 2.1.7. General Assessment of Functioning (GAF)

The GAF is an integral part of the multiaxial psychiatric diagnostic system, as outlined in the Diagnostic and Statistical Manual IV edition (DSM-IV) [27] and considered before the publication of the DSM-5 [28]. The purpose of this scale was to supplement existing data regarding the categorical diagnosis by adding information on general functioning in major areas of interest (e.g., social, work, school). Scores range from 100 to 1, with higher scores indicating a higher level of functioning.

#### 2.1.8. Statistical Analysis

All statistical analyses were performed with R Studio 4.4.0 version [29] and JASP 0.16.3.0 [30]. The Kruskal–Wallis test was employed to analyze quantitative values, and the χ^2^ test was used for qualitative variables. We used the Shapiro–Wilk Test to evaluate the normality of data distribution. We probed the association of PID-5-defined traits and domains with the risk of suicide, scored with the BPRS-24 item 4 initially, with a Spearman correlation coefficient. Thereafter, we further corrected the association for the effects of possible confounders represented by the CGI-S at recruitment, age at the time of recruitment, and sex assigned at birth within an ordinal logistic model by systematically substituting each PID-5 defined domain and traits in the described model. We further explored the potential association of ERC-R, CECA.Q, and DES-II main subscales in an ordinal logistic model to evaluate whether there might have been any association between these elements and BPRS-24 item 4. We corrected the obtained *p*-values for the effect of multiple comparisons with Holm’s method. Considering our design, we have 70% power to detect a large effect size with an odds ratio of 3.0 or higher.

## 3. Results

We identified 561 individuals who were considered potentially viable candidates for the present project. From this group, we recruited a convenience sample comprising 137 individuals receiving care from our service. Table 2 summarises reasons for exclusion.

Table 3 summarises the main sociodemographic and clinical features of the sample.

We found marginally significant associations of the BPRS-24 item 4 with the traits of suspiciousness (Spearman = 0.178, *p*-value = 0.042), perseverance (Spearman = 0.179, *p*-value = 0.040), rigid perfectionism (Spearman = 0.179, *p*-value = 0.040), reduced affectivity (Spearman = 0.180, *p*-value = 0.038), eccentricity (Spearman = 0.171, *p*-value = 0.050), grandiosity (Spearman = 0.171, *p*-value = 0.050), impulsivity (Spearman = 0.181, *p*-value = 0.038), and the disinhibition domain (Spearman = 0.200, *p*-value = 0.020) using the Pearson correlation coefficient. Furthermore, we observed more robust associations for the traits of anhedonia (Spearman = 0.227, *p*-value = 0.009), depressivity (Spearman = 0.258, *p*-value = 0.003), intimacy avoidance (Spearman = 0.291, *p*-value = 0.001), and the detachment domain (Spearman = 0.301, *p*-value = <0.001). We found no association between the DES total score, or any subscale or ERC-R, and the fourth item of the BRPS-24. We found marginally significant associations for the CECA.Q-defined mother psychological abuse subscale (Spearman = 0.200, *p*-value = 0.019) and for mother neglect (Spearman = 0.214, *p*-value = 0.012). To evaluate the robustness of our findings, we explored the association with BPRS-24-defined suicide risk and PID-5-defined traits and domains using an ordinal logistic regression with suicide risk as the dependent variable, correcting the model for the CGI-S, age at the time of recruitment, and sex assigned at birth (Figure 2).

In our sample, anhedonia showed no independent association with elevated suicide risk after adjusting for age, gender, and current CGI screening (Panel A: OR = 1.09, 95% CI 0.75–1.59, *p* = 0.638). By contrast, each one-unit increase on the PID-5 depressivity scale was associated with 1.80-fold higher odds of BPRS-4–defined elevated suicide risk (Panel B: OR = 1.80, 95% CI 1.23–2.63, *p* = 0.002, *p*-value adjusted Holm’s method 0.012). When adding BPRS-24 item 3 on depression to the same model, the association between BPRS-24 item 4 and PID-5 detachment and depressivity was no longer significant, suggesting that in our sample, there was a significant association between BPRS-24-defined depression and suicidal ideation and that the former may mediate this association. The variance inflation factor (VIF) of 2.2 for the possible collinearity between BPRS-24 items 3 and 4 supports the viability of using BPRS-3 as a covariate, indicating some collinearity but within acceptable levels. Similarly, the VIF coefficient for PID-5 depressivity and BPRS-3 depression severity shows no significant collinearity and is equal to VIF = 1.3, which is well within the acceptable levels for analyses. We further explored with structural equation modeling (SEM) the association between the selected PID-5 traits and domains with BPRS-4 and the potential mediation of BPRS-3 depression. The SEM results indicate that BPRS-3 depression fully mediates the association between depressivity and BPRS-4 suicide risk in our sample (Figure 3). The indirect effect was significant (beta = 0.276, 95% CI bootstrap < 0.001), while the direct effect of depressivity on suicidality was non-significant (beta = −0.050, *p* = 0.625), suggesting that the effect of depressivity on suicide risk was fully mediated by state depression severity in our sample, with up to 24% of the variance of BPRS-3 explained by PID-5 depressivity. However, this phenomenon may also represent an inflation of depressivity led by an acute depressive state. SEM modeling for anhedonia and detachment was not significant.

We found no evidence for an association for any of the CECA.Q subscales, ECR-R, or DES-II score with BPRS-24-defined suicide risk or for the remaining traits used to calculate the detachment domain (other than anhedonia, i.e., withdrawal and intimacy avoidance traits) in the same model. Considering the possible impact of ongoing pharmacological treatment in the observed associations, we further explored in our model the influence of dichotomized atypical antipsychotics, typical antipsychotics, mood stabilizers and antidepressants, anxiolytic therapy, individual psychotherapy, group psychotherapy, and individual rehabilitation. As expected, considering the limited sample size, when adding these variables to the model, no significant association survived *p*-value correction for multiple comparisons. However, both mood stabilizers and anxiolytics present a numerical tendency for lower BPRS-4-defined suicide risk (OR = 0.30 and 0.35 and adjusted *p*-values of 0.126 and 0.38, respectively). The likelihood ratio between the simpler model with no treatment type variables as compared with the one comprising this adjustment suggests a better fit for including treatment variables (LR stat = 18.12, df = 8, *p* = 0.0204). However, the Akaike Information Criterion shows no benefit for adding therapy variables, whilst the Bayesian Information Criterion suggests that the increase in fit is modest with their addition and sensitive to penalization for model complexity (BIC with therapies 279.35 vs. no therapies 258.40). Therefore, we considered resorting to applying the simpler model, especially considering the limited sample size.

## 4. Discussion

Personality traits have been demonstrated to be persistent over time [31] and have been extensively studied in association with suicidal behaviors [13,32,33] and non-suicidal self-harming [13,34]. Previous peer-reviewed papers explored the possible persistence of PID-5 personality traits and depression severity, finding a significant association for the depressivity and anhedonia traits [35]. Somma et al. explored the association of selected PID-5 traits in adolescents and also found an association for submissiveness, depressivity, and anhedonia with lifetime life-threatening suicide attempts but corrected only for the presence of mood disorders and not for depression severity [36,37]. Past reports suggest that even among non-clinical populations, the PID-5-defined domain of detachment could be associated with both suicidal ideation and attempts, with the domains of disinhibition and psychoticism being associated with suicidal behavior and negative affect only to suicide ideation [38]. Non-suicidal self-harming and suicidal ideation have also been reported in association with higher levels of PID-5-defined negative affectivity, detachment, antagonism, and psychoticism in people accessing mental health services [34]. Maladaptive personality traits have been reported in association with suicide ideation, even when correcting for depression severity [39,40]. However, to the best of our knowledge, no specific paper has addressed the association of PID-5 traits with suicide behaviours when controlling for depression severity. Table 4 summarizes some of the most pertinent reports in the field.

Here, we explored the potential association of current BPRS-24-defined suicidal ideation with PID-5-defined personality domains in clinical populations, correcting for concomitant depression severity with BPRS-3, practiced therapies, and the possible interaction of early-life traumas, attachment styles, and dissociative experiences. Contrary to our expectations, we did not find evidence for the association with suicidal ideation with any of the CECA.Q-defined early-life traumas, attachment styles as per the ERC-R, or DES-II dissociative experiences. The general consensus in the literature is that early-life traumas may be associated with a higher risk for suicide ideation, albeit with small possible differences, depending on the most significant form of trauma reported [42,43]. Bach et al. report that in a convenience sample of 124 outpatients, the cross-sectional association between childhood trauma defined according to the Childhood trauma Questionnaire and suicide behaviors appear mediated by borderline personality disorder features, with the trait facets of depressivity and perceptual dysregulation accounting for most of the effect (54% and 37%, respectively) [41]. In our study, we cannot exclude the possibility that the effects of trauma were undetectable due to limited statistical power, the relatively stronger influence of depressive symptoms captured in our model, or constraints related to the selected instruments. Moreover, the inclusion criteria—favoring participants who were sufficiently clinically stable, capable of engaging with a demanding battery of tests, and willing to participate in a non-incentivized clinical study—may have further limited variability in clinical severity, potentially attenuating observable effects. The significant correlation between depressive symptoms (BPRS-3) and suicidal ideation (BPRS-4) was an anticipated finding, consistent with the established role of depression as a proximal factor in suicidality. The anchors for BPRS-3 depressive symptoms do not explicitly mention suicide ideation, despite this being frequently reported among validated depressive symptoms scales. Therefore, we added this element to the model, exploring the potential interaction between these two items. However, it is important to note that BPRS-3 and BPRS-4 items are clinician-rated and susceptible to overlapping evaluative criteria. When clinicians detect suicidal ideation, they may—consciously or unconsciously—attribute greater severity to depressive symptoms, contributing to potential bias in scoring. In contrast, the PID-5 offers a dimensional, self-administered assessment of broad personality traits, albeit with possible continuity with active psychopathological states, along with the possibility of tracking eventual changes over time and, therefore, may offer further advantage as an additional clinical tool [44]. Concerns surrounding circularity of reasoning have long plagued the field, leading to criticism surrounding, for instance, the association between major depressive disorder severity and risk of suicide. In fact, depression severity is one of the strongest correlates for suicide in patients with major depressive disorder [45], with suicide ideation and attempts representing a frequent element included in the severity assessment scales [46]. This problem may arise even when considering depression severity based on the number of criteria, as suicide ideation is also a criterion for the diagnosis. Although it is not a mandatory diagnostic criterion for major depressive disorder, it offers an additional layer of complexity in adequately addressing severity measures, as it may be part of the definition and also included in the assessment of its severity, other than also representing an outcome [47]. Notwithstanding possible concerns surrounding collinearity between PID-5 anhedonia and depressivity facets with BPRS-3 depression severity, across different analytic assessments, we found no evidence of problematic collinearity among the studied variables in our sample. Depressivity was associated with suicidal ideation, but concurrent depressive symptoms fully mediated these associations. Our findings are consistent with the previous literature in the field, showing that PID-5 depressivity correlates strongly with current depressive symptoms. An additional report described the continuity in content between depressivity as a PID-5 facet with depressive symptomatology as defined according to the Center for Epidemiologic Studies Depression Scale (CES-D), finding that PID-5 depressivity and anhedonia were the only facets to predict CES-D-defined depression severity [35]. A meta-analysis exploring the association between different symptom disorders clustered into six main categories (i.e., depressive, anxiety, addictive and compulsive disorders, substance use disorders, trauma and stress-related disorders, thought disorders) and PID-5-defined domains found evidence for a significant association with PID-5-defined domains and several of these categories, extending the possible utility of the measure beyond the Alternative Model for Personality Disorder of the Statistical Manual of Mental Disorders 5th edition [44]. Consistent with a dimensional model of psychopathology more broadly, these results may support the hypothesized relationship between core personality dysfunction and symptom disorders as postulated according to the Hierarchical Taxonomy of Psychopathology [48]. When paired with the Alternative Model of Personality Disorder criterion A for impairments in personality functioning, the PID-5 may assist in informing the overall level of psychopathology and as a method for screening a wide range of personality dysfunction prior to a more focused interview-based assessment for syndromes and symptoms, or for additional clinician-rated or self-rated scales to compliment the corresponding profiles [44].

### 4.1. Future Directions

In a non-linear passage paradigm from suicidal ideation to plan and ultimately leading to attempt [4], personality traits may represent trait-level risk factors interacting with more proximal depressive states in influencing suicide risk. Arguably, selected personality traits may represent one of the possible long-term predisposing factors in this framework, especially in light of their apparent stability over time, association with psychosocial functioning [31], and continuity with symptoms over a broad range of psychiatric disorders [44]. Psychotherapy has been reported to impact psychological traits, and it may be aptly applied in the appropriate clinical scenario to manage maladaptive personality traits in clinical samples [49,50]. Considering the relative rarity of suicides in clinical samples [51], documenting the efficacy of suicide death prevention of psychotherapeutic interventions aimed at maladaptive personality traits may represent a daunting task. However, similarly to what is typically performed in other medical fields for other rare events [3], stratifying patients’ populations based on reasonably long-term risk may enable focusing scalable and cost-effective interventions for those subjects perceived to be exposed to the highest risk [3,52,53]. Psychological capital is typically listed among the different suicide resilience attributes, with personality traits representing one of the different constituents influencing the development of suicide behaviors and the suicidal recovery process for survivors [54]. Although, in our cross-sectional analysis of PID-5 facets, we did not independently predict suicidal ideation once concurrent depressive symptoms were taken into account. We believe that PID-5 may still represent a valuable screening instrument to assess a wide range of psychopathological states in continuity rather than as distinct from symptom severity, thus allowing for accurate evaluation of personality facets in longitudinal assessments [44]. Our findings suggest a continuity between personality traits and clinical symptoms across a variety of psychiatric disorders (e.g., depression, anxiety, trauma-related disorders). This continuity highlights the potential applicability of suicide risk assessments to diverse clinical populations, provided individuals maintain sufficient clinical stability to engage meaningfully with the proposed tests. Future longitudinal studies with adequate statistical power could clarify the clinical utility and impact of such assessment tools over time. This understanding could inform preventive strategies and targeted psychotherapeutic interventions aimed at modifying maladaptive cognitive–affective patterns associated with enduring personality traits, which often present in continuity with other psychiatric conditions. Recent advances in artificial intelligence may further enhance suicide risk prediction models. Machine learning models leveraging the analysis of sociodemographic, clinical, and health data have shown some promise in early risk detection, possibly allowing for an intensification of treatment efforts when tailored for specific patient populations [55] or capitalizing on administrative data [56]. A particularly promising avenue may be represented by the use of ecological momentary assessments [57] or telephone recordings to process speech analysis [58]. Integrating dynamic personality traits, psychopathological profiles, and clinical histories into predictive models supported by recent advances in artificial intelligence may facilitate the timely identification of high-risk individuals and promote more personalized treatment protocols.

### 4.2. Limitations

Several limitations should be discussed. It is possible that by selecting subjects willing to fill in a significant number of questionnaires, we might have introduced bias in our analysis, ultimately identifying a subpopulation of subjects with current suicidal ideation and attempts that may have different features than the regular clinical population. The BPRS-24 item 4 may potentially conflate attempts with suicide ideation with plan, especially for scores equal to or higher than 5. As we did not record any other additional elements regarding suicidal behavior, we could not correct for these two different outcomes. No meaningful correction was possible for the presented results according to PID-5-defined validity scales; therefore, we cannot rule out the potential influence of response styles on self-rated dissociative experiences and early-life traumas [59]. However, our findings appear largely in line with previous reports [34,38] in the literature, possibly supporting their viability. Due to the nature of recorded data, we could not further explore the association of PID-5-defined personality domains with the self-harming SCID-II-defined borderline trait, as the number of traits was registered cumulatively with no possible way of qualitatively discerning the different traits (i.e., we could not differentiate those featuring self-harming as an SCID-IV-defined trait from the others). A single-item BPRS-3 depression severity may somewhat hinder the possibility of exploring the possible contribution of different components of the depressive experience in the association of suicide behaviours with PID-5 facets. For example, BPRS-3 depression anchors include anhedonia in its definition, with possible areas of overlap with the PID-5 anhedonia trait and virtually limiting the possibility of teasing out other elements of the depressive experience and exploring their association with the PID-5 anhedonia trait itself. Another element worth considering is the potential overlap between state and trait anhedonia and depressivity. Previous reports have suggested that state anhedonia rather than traits may be associated with suicidal ideation when correcting for depression severity [40]. In the BPRS-3 anchors for depression severity, anhedonia is specifically mentioned among the elements to be assessed, therefore substantiating possible collinearity concerns. Despite these possible criticisms, our analysis suggests that collinearity appears in our database within acceptable limits to allow for analyzing these elements in the same model. The depressivity facet, as defined according to the PID-5, is scored based on fourteen questions, of which three specifically question items regarding suicide [60]. To the best of our knowledge, only one paper probing the potential mediating role of the PID-5 traits in the association between childhood trauma and suicide ideation excluded these three items from the facet, still finding a significant mediating effect for depressivity, albeit of reduced intensity (with the mediating effect decreasing from 0.007 to 0.005 in the model) [41]. Two remaining papers included in the discussion of this paper exploring the association between suicide ideation and PID-5 traits did not apply this correction and used the standard depressivity facet [38,61]. For our analysis, we could not recalculate the depressivity facet without the three suicide items; therefore, it should be considered as a possible limitation of the present report. However, none of the cited reports addressed the severity of concurrent depression as a state and its possible continuity with depressivity as a trait, therefore corroborating the novelty of our report. Considering the extensive battery of assessments used in this project, participation required a minimum level of clinical stability. Consequently, the most impaired subjects—who might have had difficulties completing a 220-item self-rated instrument, such as the PID-5—were excluded. This affected only 18 individuals among the 561 assessed for eligibility, making it unlikely to have substantially influenced our findings. Nonetheless, this selection process may have led to the inclusion of participants with predominantly mild-to-moderate clinical severity. Similar patterns have been described in the literature, where only about one-fifth of individuals with severe mental illnesses, such as schizophrenia, are represented in clinical studies [62]. Suicide ideation and attempts represent only proxies for death by suicide. Considering that only a portion of individuals dying by suicide had contact with mental health services and that only 20% of individuals dying by suicide had a previous suicide attempt [63], it is particularly relevant to consider that these findings may be generalizable only to clinical populations and even then only to a subpopulation of individuals taking part in clinical studies. Suicide ideation and attempts should, therefore, only be considered proxies for identifying a population exposed to an increased risk of lifetime suicide risk but not overlapping with the individuals ultimately dying by suicide. The relatively limited power of our analysis would allow us only to detect a large signal for the associations observed; therefore, we cannot reliably rule out a possible type II error for the presented results.

## 5. Conclusions

We found a significant association between BRPS-24-defined current suicide ideation and the PID-5 depressivity trait, although these effects were mediated by depression severity. Contrary to our expectations, we found no significant association between current suicidal ideation and attachment styles, early-life traumas, or dissociative experiences. Our results appear largely in line with previous reports, suggesting continuity of state and trait depressive symptoms, with personality traits working as indirect contributors to suicide risk mediated by acute depressive symptoms. Longitudinal studies may help define the role of PID-5 in suicide risk stratification and in profiling psychological capital in suicide resilience.

## Figures and Tables

**Figure 1 medicina-61-01541-f001:**
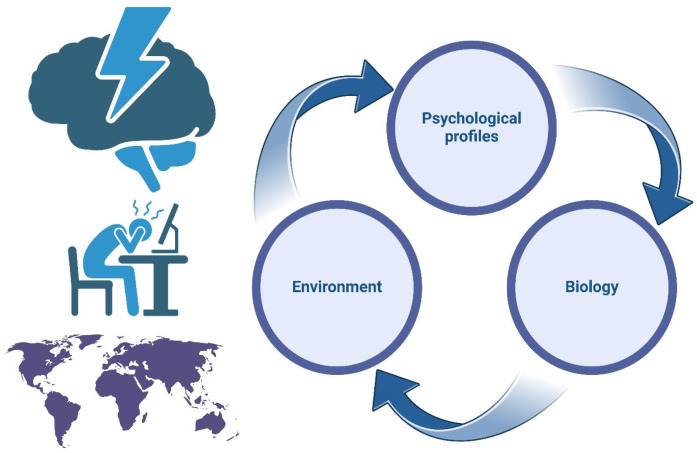
Biological, environmental, and psychological elements may influence suicide risk at any given time. Created in BioRender. Paribello, P. (2025). https://BioRender.com/jx5i370, accessed on 10 July 2025.

**Figure 2 medicina-61-01541-f002:**
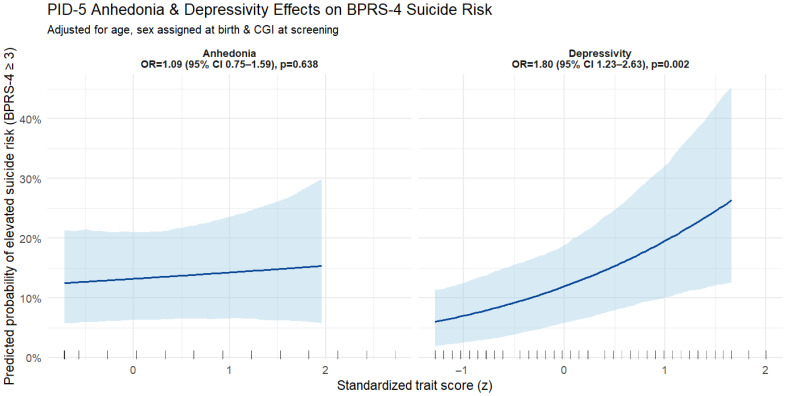
Marginal effect plot for the logistic regression model of BPRS-4 suicide risk association with anhedonia on the left and depressivity on the right of the figure. Both models were corrected for sex assigned at birth, age, and CGI-S at screening.

**Figure 3 medicina-61-01541-f003:**
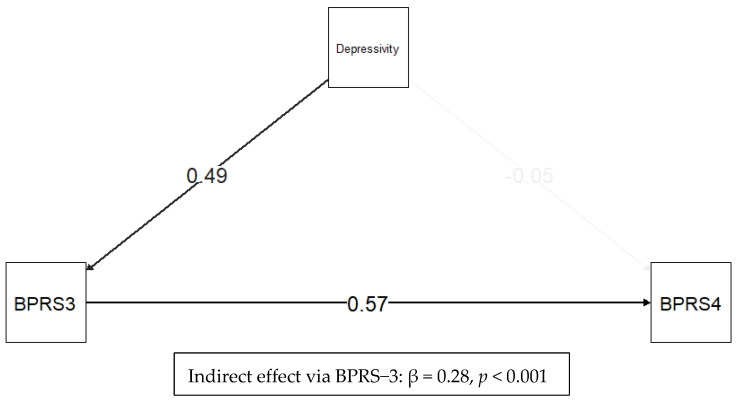
Path diagram in standard equation modeling for depressivity, BPRS-3, and BPRS-4. Arrows represent standardized path coefficients from the structural equation model. The path from depressivity to BPRS-3 depressive symptoms was significant (β = 0.49, *p* < 0.001), as was the path from BPRS-3 depressive symptoms to BPRS-4 suicide risk (β = 0.57, *p* < 0.001). The direct effect of depressivity on BPRS-4 suicide risk was not significant (β = −0.05, *p* = 0.625), indicating full mediation. The indirect effect of depressivity on BPRS-4 suicide risk via BPRS-3 depressive symptoms was, however, significant (β = 0.28, *p* < 0.001), accounting for a considerable part of the total effect (β = 0.23, *p* = 0.017).

**Table 1 medicina-61-01541-t001:** BPRS-24—item 4: suicide risk anchors.

Item 4—Suicide Risk
1. Symptom Absent
Deny suicidal ideation
2. Very Mild
Occasionally tired of living. No suicidal thoughts.
3. Mild
Occasional suicidal thoughts that do not translate into a clear decision or plan, and/or the patient often has the impression that it would be better if they were dead.
4. Moderate
Frequent suicidal thoughts without a concrete decision or established plans to end one’s life.
5. Moderately Severe
The patient has many suicidal fantasies and thinks about different ways to end their life. They may also formulate precise plans or set a specific time to do so and/or have made an impulsive suicide attempt using a non-lethal method or knowing they could be saved.
6. Severe
The patient clearly wants to end their life. They are actively seeking the right moment and means to do so and/or have carried out a serious suicide attempt, even if the method used did not completely rule out the possibility of being rescued.
7. Very Severe
The patient has a clear intention and a well-defined suicide plan (e.g., “As soon as… I will end my life by doing…”) and/or has made a suicide attempt using a method they believed to be certainly lethal or that was inherently dangerous and carried out in an isolated location.

**Table 2 medicina-61-01541-t002:** Reasons for exclusion from the present study.

Reasons for Non-Eligibility	N	% of Total (n = 561)	Relative % Among Excluded (n = 424)
No longer in care	260	46.3%	61.3%
Cognitive impairment	37	6.6%	8.7%
Clinical severity (CGI-S > 4)	18	3.2%	4.2%
Unavailable	105	18.7%	24.8%
Language difficulties	3	0.5%	0.7%
Deceased	1	0.2%	0.2%
Subtotal	424	75.6%	100%
Included in the study	137	24.4%	—

**Table 3 medicina-61-01541-t003:** Sociodemographic and clinical variables divided based on BPRS-4 suicide risk.

Variable (Median or %)	BPRS Suicide Risk ≤ 3	BPRS Suicide Risk > 3	*p*-Value *
Sex assigned at birth female—n (%)	75 (62.5)	14 (82.3)	0.108
GAF	60.0	55.0	0.163
CGI	4.0	4.0	0.219
Years of observation	2.7	3.6	0.705
Age	54.4	55.1	0.793
Education—n (%)			0.629
Primary	9 (7.5)	2 (11.7)	
Secondary	43 (35.8)	3 (17.6)	
High school	42 (35.0)	8 (47.0)	
University	26 (21.6)	4 (23.5)	
Occupation—n (%)			0.470
Working	55 (45.8)	9 (52.9)	
Homemaker	8 (6.6)	2 (11.7)	
Student	12 (10.0)	2 (11.7)	
Retired	7 (5.8)	2 (11.7)	
Unemployed	38 (31.6)	2 (11.7)	
Age of onset—(years, mean)	28.0	27.0	0.829
CECA—mother antipathy	16.0	22.0	0.117
CECA—mother neglect	15.5	19.0	0.023
CECA—mother psychological abuse	2.0	8.0	0.042
CECA—father antipathy	17.0	16.0	0.463
CECA—father neglect	19.0	20.0	0.243
CECA—father psychological abuse	2.0	4.0	0.200
CECA—role inversion	45.0	51.0	0.378
PID-5 domains and validity scales			
Negative affect	1.2	1.4	0.395
Detachment	1.0	1.4	0.104
Antagonism	0.2	0.4	0.339
Disinhibition	0.9	0.9	0.560
Psychoticism	0.6	0.0	0.165
ERC attachment styles			
Anxious attachment	65.0	65.0	0.837
Avoidant attachment	72.0	68.0	0.445
Substance use disorder	15 (12.5)	1 (5.8)	0.427
Personality disorder (at least one)	42 (35)	11 (64.7)	0.019
BPRS-24 item 3: depression severity	2	4	<0.001
Current mood disorder diagnosis	54 (45)	12 (70)	0.049
Mood stabilizer	5 (62.5)	94 (72.9)	0.819
Antidepressant	3 (37.5)	50 (38.8)	1.000
Typical antipsychotic	8 (100)	124 (96.1)	1.000
Atypical antipsychotic	5 (62.5)	99 (76.7)	0.625
Anxiolytic/hypnotic	3 (37.5)	62 (48.1)	0.829
Psychotherapy/rehabilitation	4 (50.0)	79 (61.2)	0.796
Individual psychotherapy	4 (50.0)	84 (65.1)	0.627
Group psychotherapy	6 (75.0)	120 (93.0)	0.250

Abbreviations: * We employed Student’s *t*-test for normally distributed variables, the Kruskal–Wallis test for non-normally distributed ones, and χ^2^ for categorical variables.

**Table 4 medicina-61-01541-t004:** Summary of the most significant papers probing the association between PID-5 and depression or suicide behaviors. Abbreviations: Center for Epidemiologic Studies Depression Scale (CES-D).

Study/Year	Sample Type	Controlled for Depression Severity?	Depressivity—Suicidality Association
Somma et al. (2016) [36]	Adolescent inpatients	Mood diagnosis only	Significant (depressivity, anhedonia, submissiveness)
De Salve et al. (2023) [34]	Young adults (general + clinical)	No	Significant (negative affectivity, detachment)
Aboul-Ata et al. (2023) [38]	College-aged young adults	No	Significant (depressivity, detachment, psychoticism)
Gonçalves et al. (2022) [35]	Portuguese community adults	No	Not specifically addressing suicidality—found that depressivity and anhedonia were associated with CES-D depression severity
Bach & Fjeldsted (2017) [41]	In- and outpatient mental health services	No	Negative affectivity traits (incl. depressivity) mediate the trauma–suicidality link. Addressed potential circularity by excluding the three PID-5 items surrounding suicide, still finding a significant association

## Data Availability

Data is available upon reasonable request to the study authors.

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
