# Peer review of "Association Between Current Suicidal Ideation and Personality Traits: Analysis of the Personality Inventory for DSM-5 in a Community Mental Health Sample"

_medicina, 2025, doi:10.3390/medicina61091541_

Round 1

Reviewer 1 Report

Comments and Suggestions for Authors

This is a clinical selected case-series aiming at studying the potential association between suicidal ideation and domains in a personality scale (PID-5).The aim is stated in the abstracts ”The PID-5 may serve as a clinical tool to support the identification of individuals at long-term risk of suicidality or for targeting specific populations for tailored interventions” (p 1, 35)

Comment 1.

I suggest aim is stated at the end of introduction.

Sample

The original potential sample consisted of 526 subjects from outpatient psychiatric care. 137 were recruited, about 25%. Excluded were subjects with cognitive and language problems as well as those with CGI severity score higher than 5.

Comment 2.

 It is clear that subjects with more severe illness were excluded. The noted GAF-scale of 60 and 55 is above the GAF-threshold for clinical negative status (”cutoff” is 50) As suicidality is common in more severe cases this might lead to exclusion of many subjects with suicidality. In order to clarify this it is necessary to present data on the CGI-reason excluded group. How many? Clinical characteristics? What was the suicidality score of that sample? GAF-scale? Etc.

Comment 3.

The paper aims at studying personality. But in this sample personality is under the influence of ongoing pychiatric care with medication and other interventions. It is a sample under influence that probably affects the delienation of personality. This is problematic and does make it impossible to draw conclusions generalized to personality traits in subjects outside of ongoing psychiatric treatment.

Comment 4.

There was no presentation of ongoing medication i the sample. This is of great importance as such medication migh have an effect both on personality and suicidality. For instance do SSRIs carry an increase of anhedonia and an increased risk of suicidal ideation and neuroleptics affect cognition and emotions. Ongoing medication is of such importance that it should be presented both in the excluded and the included sample. In order to study the personality and suicidality not under influence of psychiatric drugs, those subjects without such medication culd have been selected.

References

Comment 5.

Reference 5, which is given importance, is cited as ”…only 20% of suicide decendents being engaged in psychiatric care”. On reading it turns out that that figure comes from another paper (Liu et al. Surveillance summaries/May 26,2023/75(5),1-38.) There decendents of suicide are analysed and about half of subjects had psychiatric problems, 32 % had previous and 24% ongoing psychiatric care.

The substance of ref 5 is the classification of subgroups of which the largest is physical health care problems (32%) The other four groups were polysubstance abuse, crisis in relation to work etc, mental problems and mental problems with substance abuse. The logic from this paper is that we should look at the subgroups for different prevention strategies.

With the largest group, problems with physical health probably most subjects had contact with health care, althouhg not psychiatric care although all doctors have a psychiatric training. Thus also somatic doctors could be open to probe suicidality in their patients.

For the present paper to use ref 5 should present the findings of that paper which are of great importance in understanding the  complexity with subgroups regarding suicide.

Bias of circularity

Comment 6.

The paper states: ”We found statistically isgnificant associations between suicidal ideation and the PID-5 trait of depressivity” (p 1:29)

 In the PID-5 trait of depressivity there are three items on suicidality (81, 119,178) To state an association between suicidal ideation and PID-5 depression is thus bias of circularity which is not acceptable.

Results

Comment 7.

The paper states ”…we observed more robust associations for the traits of anhedonia…and for the detachment domain” (p7:19)

These traits are also common effects of antidepressants and neuroleptics. Thus this must be noted as an alternative explanation and preferably the medication status of the subjects should be presented.

Discussion

The paper states: ”Contrary to our expectations we did not find evidence for the association with suicidal ideation wit any of the CECA.Q-defined early life trumas…or DES-II dissociative experiences” (p9:24).

Comment 8.

The more severe clinical cases were excluded from the sample. Severe childhood trauma might be associated with those more severe cases.Thus it would be correct to comment on that and only say something like ”in less severe clinical caes there was no association…” As early childhood traua id discussed it woul have been appropriate with a reference on that, as discarding of such association is part of the discussion.

The paper states: ”Considering the growing body of evidence suggesting the potential  association  of PID-5-defined personality domains with suicidal behaviours…”

Comment 9.

See comment 5 on bias of circularity.

The paper states: ”…it is reasonable to hypothesize that in the future this measure and the corresponding brief forms could be adopted among as possible ancillary outcome measures for psychotherapeutic interventions aiming at adressing some of the potential elements associated with greater risk for suicidal ideation in clinical samples”

Comment 10

This is a vague conclusion and does not have a robust base from the results. I would suggest a more clearcut message like for instance ” This study does not support the use of PID-5 in suicide prevention”

Author Response

This is a clinical selected case-series aiming at studying the potential association between suicidal ideation and domains in a personality scale (PID-5).The aim is stated in the abstracts ”The PID-5 may serve as a clinical tool to support the identification of individuals at long-term risk of suicidality or for targeting specific populations for tailored interventions” (p 1, 35)

Comment 1.

I suggest aim is stated at the end of introduction.
-R we edited the draft to reflect the required changes

Sample

The original potential sample consisted of 526 subjects from outpatient psychiatric care. 137 were recruited, about 25%. Excluded were subjects with cognitive and language problems as well as those with CGI severity score higher than 5.

-R Thank you for your comment. We added the relative reasons for study esclusion so that now our report is STROBE compliant. Ultimately, only 3.2% of the overall sample of individuals considered for inclusion has a CGI = or > 4. The number reported of 561 reflects the total number of individuals considered for inclusion, including those who refused to participate. As is often the case in clinical studies, individuals receiving care and willing to complete multiple assessment instruments represent only a subpopulation of the overall clinical population. Since those who declined participation did not provide consent, no clinical data could be collected from them, and we are therefore unable to provide any information regarding this group. Completing several instruments—including a 220-item self-report scale—requires a minimum level of clinical stability. It is therefore likely that the most severely impaired patients were among those who refused to participate, and this limitation has been acknowledged in the present study.

 Comment 2.

 It is clear that subjects with more severe illness were excluded. The noted GAF-scale of 60 and 55 is above the GAF-threshold for clinical negative status (”cutoff” is 50) As suicidality is common in more severe cases this might lead to exclusion of many subjects with suicidality. In order to clarify this it is necessary to present data on the CGI-reason excluded group. How many? Clinical characteristics? What was the suicidality score of that sample? GAF-scale? Etc.
-R Thank you for your comment. Unfortunately, excluded patients did not consent to participate and therefore, no clinical data was recorded. It is very plausible that among this population there could be more severely affected subjects. However, considering that only 3.2% of excluded patients (n=18) were excluded for their CGI-S severity,  this element did not represent a significant reason for exclusion in our sample.

Comment 3.

The paper aims at studying personality. But in this sample personality is under the influence of ongoing pychiatric care with medication and other interventions. It is a sample under influence that probably affects the delienation of personality. This is problematic and does make it impossible to draw conclusions generalized to personality traits in subjects outside of ongoing psychiatric treatment.

-R Thank you for your feedback. Considering the naturalistic nature of this study, excluding the effect of ongoing treatment for this relatively small, convenience sample may represent a daunting task. We added further data on the practised treatment and on the use of dichotomised therapy in the logistic ordinal model applied. Specific analysis focusing on the type of singular molecules or their doses would probably not be feasible under these circumstances

Comment 4.

There was no presentation of ongoing medication i the sample. This is of great importance as such medication migh have an effect both on personality and suicidality. For instance do SSRIs carry an increase of anhedonia and an increased risk of suicidal ideation and neuroleptics affect cognition and emotions. Ongoing medication is of such importance that it should be presented both in the excluded and the included sample. In order to study the personality and suicidality not under influence of psychiatric drugs, those subjects without such medication culd have been selected.

-R Thank you for your feedback. Considering the naturalistic nature of this study, excluding the effect of ongoing treatment for this relatively small, convenience sample may represent a daunting task. We added further data on the practiced treatment and on the use of dichotomized therapy in the logistic ordinal model applied.

References

Comment 5.

Reference 5, which is given importance, is cited as ”…only 20% of suicide decendents being engaged in psychiatric care”. On reading it turns out that that figure comes from another paper (Liu et al. Surveillance summaries/May 26,2023/75(5),1-38.) There decendents of suicide are analysed and about half of subjects had psychiatric problems, 32 % had previous and 24% ongoing psychiatric care.

The substance of ref 5 is the classification of subgroups of which the largest is physical health care problems (32%) The other four groups were polysubstance abuse, crisis in relation to work etc, mental problems and mental problems with substance abuse. The logic from this paper is that we should look at the subgroups for different prevention strategies.

With the largest group, problems with physical health probably most subjects had contact with health care, althouhg not psychiatric care although all doctors have a psychiatric training. Thus also somatic doctors could be open to probe suicidality in their patients.

For the present paper to use ref 5 should present the findings of that paper which are of great importance in understanding the  complexity with subgroups regarding suicide.

-R thank you for your feedback. We edited the paper as follows”. Evidence from violent death registry suggests that a large and growing proportion of individuals dying by suicide (31%) featured prominent physical health concerns and minimal psychotropic medication usage as compared with other suicide profiles.” Also, the citation [5] was adjusted to reflect the requested changes.

Bias of circularity

Comment 6.

The paper states: ”We found statistically isgnificant associations between suicidal ideation and the PID-5 trait of depressivity” (p 1:29)

 In the PID-5 trait of depressivity there are three items on suicidality (81, 119,178) To state an association between suicidal ideation and PID-5 depression is thus bias of circularity which is not acceptable.

R- thank you for your feedback. We edited the paper as follows “ Concerns surrounding circularity of reasoning have long plagued the field, leading to criticism surrounding, for instance, the association between major depressive disorder severity and risk of suicide. In fact, depression severity is one of the strongest correlates for suicide in patients with major depressive disorder [45], with suicide ideation and attempts representing a frequent element included in the severity assessment scales [46]. This problem may arise even when considering depression severity based on the number of criteria, as suicide ideation is also a criterion for the diagnosis. Although it is not a mandatory diagnostic criterion for major depressive disorder, it offers an additional layer of complexity in adequately addressing severity measures, as it may be part of the definition and also included in the assessment of its severity, other than also representing an outcome [47]”
And also:
“Depressivity facet, as defined according to the PID-5, is scored based on 14 questions, of which three specifically question items regarding suicide [47]. To the best of our knowledge, only one paper probing the potential association between suicide ideation and PID-5 Depressivity excluded these three items from the facet, still finding a signif-icant association between the two, albeit reduced [48], with two remaining papers not applying this correction and using the standard Depressivity facet [37,49]. For our analysis, we could not recalculate the Depressivity facet without the three suicide items; therefore, it should be considered as a possible limitation of the present report. However, none of these addressed the severity of concurrent depression as a state as a possible confounder for this association, therefore corroborating the novelty of our re-port.”

Results

Comment 7.

The paper states ”…we observed more robust associations for the traits of anhedonia…and for the detachment domain” (p7:19)

These traits are also common effects of antidepressants and neuroleptics. Thus this must be noted as an alternative explanation and preferably the medication status of the subjects should be presented.

R- thank you for your feedback. We edited the paper as follows: “Considering the possible impact of ongoing pharmacological treatment in the observed associations, we further explored in our model the influence of dichotomised atypical antipsychotics, typical antipsychotics, mood stabilisers and antidepressant, anxiolytics therapy, individual psychotherapy, group psychotherapy and individual rehabilita-tion. As expected, considering the limited sample size, when adding these variables to the model, no significant association survives p-value correction for multiple compar-isons. However, both mood stabilisers and anxiolytics present a numerical tendency for lower BPRS-4-defined suicide risk ( OR = 0.30 and 0.35 and adjusted p-values of 0.126 and 0.38, respectively). The likelihood ratio between the simpler model with no treatment type variables as compared with the one comprising this adjustment suggests a better fit for including treatment variables (LR stat = 18.12, df = 8, p = 0.0204). The Akaike Information Criterion shows no benefit for adding therapy variables and the Bayesian Information criteria suggests that the increase in fit is modest with their ad-dition and sensitive to penalisation for model complexity (BIC with therapies 279.35 vs no therapies 258.40). Therefore, we considered resorting to applying the simpler model, especially considering the limited sample for the presented analyses.”

Discussion

The paper states: ”Contrary to our expectations we did not find evidence for the association with suicidal ideation wit any of the CECA.Q-defined early life trumas…or DES-II dissociative experiences” (p9:24).

Comment 8.

The more severe clinical cases were excluded from the sample. Severe childhood trauma might be associated with those more severe cases.Thus it would be correct to comment on that and only say something like ”in less severe clinical caes there was no association…” As early childhood traua id discussed it woul have been appropriate with a reference on that, as discarding of such association is part of the discussion.

-R Thank you for your feedback. Considering that only 18 individuals among the 424 subjects considered for inclusion were excluded because of clinical severity, this is unlikely to have played a role. However, since the selection of individuals with a minimum clinical stability is necessary to identify individuals that will fill in the tests, we added the following passage to the limitation section “Considering the extensive battery of assessments used in this project, participation required a minimum level of clinical stability. Consequently, the most impaired subjects—who might have had difficulties completing a 220-item self-rated instrument such as the PID-5—were excluded. This affected only 18 individuals among the 561 assessed for eligibility, making it unlikely to have substantially influenced our findings. Nonetheless, this selection process may have led to the inclusion of participants with predominantly mild-to-moderate clinical severity. Similar patterns have been described in the literature, where only about one-fifth of individuals with severe mental illnesses, such as schizophrenia, are represented in clinical studies.”
And in the discussion section “The general consensus in the literature is that early life traumas may be associated with a higher risk for suicide ideation, albeit with small possible differences depending on the most significant form of trauma reported [42,43]. Bach et al report that in a convenience sample of 124 outpatients, the cross-sectional association between child-hood trauma defined according to the Childhood trauma Questionnaire and suicide behaviors appear mediated by borderline personality disorder features, with the trait facets of Depressivity and Perceptual Dysregulation accounting for most of the effect (54% and 37%, respectively) [41]. In our study, we cannot exclude the possibility that the effects of trauma were undetectable due to limited statistical power, the relatively stronger influence of depressive symptoms captured in our model, or constraints re-lated to the selected instruments. Moreover, the inclusion criteria—favoring partici-pants who were sufficiently clinically stable, capable of engaging with a demanding battery of tests, and willing to participate in a non-incentivized clinical study—may have further limited variability in clinical severity, potentially attenuating observable effects.”

The paper states: ”Considering the growing body of evidence suggesting the potential  association  of PID-5-defined personality domains with suicidal behaviours…”

Comment 9.

See comment 5 on bias of circularity.

-R Thank you for your feedback- we added the following passage to the limitation section
“Depressivity facet, as defined according to the PID-5, is scored based on 14 questions, of which three specifically question items regarding suicide [56]. To the best of our knowledge, only one paper probing the potential mediating role of the PID-5 traits in the association between childhood trauma and suicide ideation excluded these three items from the facet, still finding a significant mediating effect for Depressivity, albeit of reduced intensity (with the mediating effect decreasing from .007 to .005 in the model) [41]. Two remaining papers included in the discussion of this paper exploring the association between suicide ideation and PID-5 traits did not apply this correction and used the standard Depressivity facet [38,57]. For our analysis, we could not recal-culate the Depressivity facet without the three suicide items; therefore, it should be considered as a possible limitation of the present report. However, none of the cited reports addressed the severity of concurrent depression as a state and its possible con-tinuity with Depressivity as a trait, therefore corroborating the novelty of our report.”

The paper states: ”…it is reasonable to hypothesize that in the future this measure and the corresponding brief forms could be adopted among as possible ancillary outcome measures for psychotherapeutic interventions aiming at adressing some of the potential elements associated with greater risk for suicidal ideation in clinical samples”

 Comment 10

This is a vague conclusion and does not have a robust base from the results. I would suggest a more clearcut message like for instance ” This study does not support the use of PID-5 in suicide prevention”

-R Thank you for your feedback. We edited the paper as follows
“A meta-analysis exploring the association between different symptom disorders clustered into six main categories (i.e., depressive, anxiety, addictive and compulsive disorders, substance use disorders, trauma and stress-realted disorder, thought disoders) and PID-5-defined domains, founding evidence for a significant association with PID-5-defined domains and several of these categories, extending the possible utility of the measure beyond the Alternative Model for Personality Disorder of the Statistical Manual of Mental Disorders 5th edition [44]. Consistent with a dimensional model of psychopathology more broadly, these results may support the hypothesised relationship between core personality dysfunction and symptom disorders as postulated according to the Hierarchical Taxonomy of Psychopathology [48]. When paired with the Alternative Model of Personality Disorder criterion A for impairments in personality functioning, the PID-5 may assist in informing the overall level of psychopathology and as a method for screening a wide range of personality dysfunction prior to a more focused interview-based assessment for syndromes and symptoms, or for additional clinician rated or self-rated scales to compliment the corresponding profiles [44].”
We also added a “Future Direction” section concluding that “Although in our cross‑sectional analysis PID‑5 facets did not independently predict suicidal ideation once concurrent depressive symptoms were taken into account, we believe that PID-5 may still represent a valuable screening instrument to assess a wide range of psychopathological states in continuity rather than as distinct from symptom severity, thus allowing for accurate evaluation of personality facets in longitudinal as-sessments [44]. Our findings suggest a continuity between personality traits and clini-cal symptoms across a variety of psychiatric disorders (e.g., depression, anxiety, trau-ma-related disorders). This continuity highlights the potential applicability of suicide risk assessments to diverse clinical populations, provided individuals maintain suffi-cient clinical stability to engage meaningfully with the proposed tests. Future longitu-dinal studies with adequate statistical power could clarify the clinical utility and im-pact of such assessment tools over time. This understanding could inform preventive strategies and targeted psychotherapeutic interventions aimed at modifying maladap-tive cognitive-affective patterns associated with enduring personality traits, which of-ten present in continuity with other psychiatric conditions. Incorporating artificial in-telligence into future suicide risk prediction models could enhance the accuracy and reliability of clinical assessments. Moreover, integrating dynamic personality traits, psychopathological profiles, and clinical histories into predictive models may facilitate the timely identification of high-risk individuals and promote more personalised treatment protocols.”

Reviewer 2 Report

Comments and Suggestions for Authors

I have reviewed the manuscript entitled “Association between current suicidal ideation and personality traits: analysis of the Personality Inventory for DSM-5 in a community mental health sample” as a psychiatrist. The topic is highly relevant and clinically significant, particularly in the context of understanding risk factors for suicidality in outpatient psychiatric populations. The use of validated instruments such as the PID-5, BPRS-24, DES-II, CECA.Q, and ECR-R strengthens the study’s methodological foundation. However, there are several concerns that need to be addressed to improve the scientific rigor and clarity of the manuscript.The study’s sample size of 137 participants is relatively small for the number and complexity of statistical analyses performed. I recommend conducting and reporting a priori power analysis using a software such as G*Power to support the adequacy of the sample and the interpretability of non-significant results. Without this, there is a risk of type II error in the associations explored. Furthermore, although the study uses BPRS-24 item 4 to assess suicidal ideation, the item conflates suicidal thoughts with suicidal behaviors at higher scores. This weakens the construct validity of the dependent variable and should be explicitly discussed. Using a more specific suicidality scale or at least supplementing with an additional measure would improve the precision of outcome measurement in future research.The manuscript would benefit from the addition of a section discussing the integration of artificial intelligence and digital tools in suicide risk prediction. Recent advances in machine learning applied to mental health assessment offer innovative pathways for identifying high-risk individuals using personality traits and clinical data. Including a brief overview of this emerging area would improve the study’s relevance and position it within contemporary psychiatric research. Additionally, although a limitations paragraph is included, it should be expanded to transparently address sample representativeness, the retrospective nature of the data, the reliance on self-report measures, and the potential for confounding variables. A clear “Future Directions” paragraph should also be added, outlining recommendations such as the need for longitudinal studies, use of digital tools, and validation in larger samples.The authors are advised to review and revise the use of abbreviations throughout the manuscript. Some abbreviations (e.g., ECR-R, CECA.Q) are not clearly defined upon first use or are used inconsistently, which may hinder reader comprehension. The clarity of the figures, particularly Figure 2, should also be improved. Axis labels, statistical annotations, and readability should be enhanced to better communicate the findings. The inclusion of a table comparing the study’s findings with previous literature would help contextualize the results and strengthen the discussion. Additionally, although data availability is noted upon request, I recommend that the authors consider sharing de-identified datasets and analysis code (e.g., via GitHub or OSF) to promote transparency and reproducibility.In conclusion, while the manuscript addresses an important area of psychiatric research and provides meaningful insights into the association between personality traits and suicidality, it requires substantial revisions to meet publication standards. 

Author Response

I have reviewed the manuscript entitled “Association between current suicidal ideation and personality traits: analysis of the Personality Inventory for DSM-5 in a community mental health sample” as a psychiatrist. The topic is highly relevant and clinically significant, particularly in the context of understanding risk factors for suicidality in outpatient psychiatric populations. The use of validated instruments such as the PID-5, BPRS-24, DES-II, CECA.Q, and ECR-R strengthens the study’s methodological foundation. However, there are several concerns that need to be addressed to improve the scientific rigor and clarity of the manuscript.The study’s sample size of 137 participants is relatively small for the number and complexity of statistical analyses performed. I recommend conducting and reporting a priori power analysis using a software such as G*Power to support the adequacy of the sample and the interpretability of non-significant results. Without this, there is a risk of type II error in the associations explored.
-R we added a power analysis estimate to reflect the required changes

 Furthermore, although the study uses BPRS-24 item 4 to assess suicidal ideation, the item conflates suicidal thoughts with suicidal behaviors at higher scores. This weakens the construct validity of the dependent variable and should be explicitly discussed. Using a more specific suicidality scale or at least supplementing with an additional measure would improve the precision of outcome measurement in future research.

-R Thank you for your feedback. Unfortunately, no additional measure on suicide was available. We added a discussion surrounding limitations for the applied measures.

The manuscript would benefit from the addition of a section discussing the integration of artificial intelligence and digital tools in suicide risk prediction. Recent advances in machine learning applied to mental health assessment offer innovative pathways for identifying high-risk individuals using personality traits and clinical data. Including a brief overview of this emerging area would improve the study’s relevance and position it within contemporary psychiatric research.

-R Thank you for your feedback. We added the following passage: “Recent advances in artificial intelligence may further enhance suicide risk prediction models. Machine learning models leveraging the analysis of sociodemographic, clinical and health data have shown some promise in early risk detection, possibly allowing for an intensification of treatment efforts when tailored for specific patient populations [55] or capitalizing on administrative data [56]. A particularly promising avenue may be represented by the use of ecological momentary assessments [57]or telephone recordings to process speech analysis [58]. Integrating dynamic personality traits, psychopathological profiles, and clinical histories into predictive models supported by recent advances in artificial intelligence may facilitate the timely identification of high-risk individuals and promote more personalised treatment protocols. “

Additionally, although a limitations paragraph is included, it should be expanded to transparently address sample representativeness, the retrospective nature of the data, the reliance on self-report measures, and the potential for confounding variables.

-R Thank you for your feedback. The limitation paragraph was further expanded to reflect the required changes

A clear “Future Directions” paragraph should also be added, outlining recommendations such as the need for longitudinal studies, use of digital tools, and validation in larger samples.

-R an additional paragraph titled future directions discuss this elements

The authors are advised to review and revise the use of abbreviations throughout the manuscript. Some abbreviations (e.g., ECR-R, CECA.Q) are not clearly defined upon first use or are used inconsistently, which may hinder reader comprehension.

-R Thank you for your feedback. An additional table was added for the most frequently used abbreviations across the manuscript

The clarity of the figures, particularly Figure 2, should also be improved. Axis labels, statistical annotations, and readability should be enhanced to better communicate the findings.

-R Figure 2 was modified to reflect the required changes

The inclusion of a table comparing the study’s findings with previous literature would help contextualize the results and strengthen the discussion.

-R An additional table summarizing the results for previous reports in the field was added

Additionally, although data availability is noted upon request, I recommend that the authors consider sharing de-identified datasets and analysis code (e.g., via GitHub or OSF) to promote transparency and reproducibility.

-R thank you for your feedback. Unfortunately, due to limitations imposed from our institution we cannot share the dataset on public repository

In conclusion, while the manuscript addresses an important area of psychiatric research and provides meaningful insights into the association between personality traits and suicidality, it requires substantial revisions to meet publication standards. 

-R thank you for your feedback. As a result of your insight, we believe that the manuscript was significantly improved and we hope it might satisfy your requirements for publication

Round 2

Reviewer 2 Report

Comments and Suggestions for Authors

The authors have completely addressed all my comments, and I have no further concerns. Therefore, I recommend accepting the paper.